# The rs3825807 Polymorphism of ADAMTS7 as a Potential Genetic Marker for Myocardial Infarction in Slovenian Subjects with Type 2 Diabetes Mellitus

**DOI:** 10.3390/genes14020508

**Published:** 2023-02-16

**Authors:** David Petrovič, Petra Nussdorfer, Danijel Petrovič

**Affiliations:** 1Institute of Histology and Embryology, Faculty of Medicine, University of Ljubljana, Korytkova 2, 1000 Ljubljana, Slovenia; 2Laboratory for Histology and Genetics of Atherosclerosis and Microvascular Diseases, Institute of Histology and Embryology, Faculty of Medicine, University of Ljubljana, Korytkova 2, 1000 Ljubljana, Slovenia

**Keywords:** myocardial infarction, association study, gene polymorphism, ADAMTS7, genetic marker, risk factor

## Abstract

Background: A disintegrin and metalloprotease with thrombospondin motif 7 (ADAMTS-7) was reported to play a role in the migration of vascular smooth muscle cells and neointimal formation. The object of the study was to investigate the association between the rs3825807 polymorphism of ADAMTS7 and myocardial infarction among patients with type 2 diabetes mellitus in a Slovenian cohort. Methods: 1590 Slovenian patients with type 2 diabetes mellitus were enrolled in this retrospective cross-sectional case–control study. In total, 463 had a history of recent myocardial infarction, and 1127 of the subjects in the control group had no clinical signs of coronary artery disease. Genetic analysis of an rs3825807 polymorphism of ADAMTS7 was performed with logistic regression. Results: Patients with the AA genotype had a higher prevalence of myocardial infarction than those in the control group in recessive [odds ratio (OR) 1.647; confidence interval (CI) 1.120–2.407; *p* = 0.011] and co-dominant (OR 2.153; CI 1.215–3.968; *p* = 0.011) genetic models. Conclusion: We found a statistically significant association between rs3825807 and myocardial infarction in a cohort of Slovenian patients with type 2 diabetes mellitus. We report that the AA genotype might be a genetic risk factor for myocardial infarction.

## 1. Introduction

Atherosclerosis is a chronic inflammatory disease of large- and medium-sized muscular and elastic types of arteries, with a characteristic progressive accumulation of lipids, connective tissue, calcium and cell remains in the intima, and accompanying inflammation that affects the intima, media and adventitia of artery walls. It represents a leading cause of death worldwide, most commonly through coronary and cerebral ischemic disease [1]. The prevalence and the degree of atherosclerosis are closely correlated with several constitutive risk factors such as genetics, age, and gender and with many variable risk factors such as hyperlipidemia, hypertension, smoking, and diabetes [2].

The atherosclerotic process usually progresses slowly over the years and may be clinically silent until it causes more than 75% of stenosis of the coronary artery. Atherosclerotic plaques may sometimes be complicated by plaque rupture, thrombus formation, and vasospasm. These complications may lead to the complete occlusion of the coronary artery, causing an acute coronary event (unstable angina, acute myocardial infarction, sudden cardiac death) [3]. 

The underlying pathogenetic mechanism of plaque growth and plaque rupture is enhanced inflammatory infiltration and the enhanced activity of inflammatory cells within the plaque. Activated macrophages play an important role in plaque progression and plaque rupture. They cause endothelial and smooth muscle cell death by apoptosis. Moreover, activated macrophages produce various enzymes such as matrix metalloproteinases (MMPs) that cause the degradation of the connective tissue matrix (collagen, etc.) [3,4]. 

An important characteristic of unstable or rupture-prone plaque is a thin fibrous cap and inflammatory cells (macrophages and lymphocytes) in the shoulder region of the fibrous cap [5]. Activated macrophages with their enzymes (MMPs) play a crucial role in the degradation of the extracellular matrix (i.e., collagen) of the fibrous cap. Another histological characteristic of unstable or rupture-prone plaque is a large lipid-rich necrotic core. It was reported that a larger lipid-rich core produces greater pressure on overlying fibrous caps and higher amounts of prothrombotic material [3,4]. Due to all these processes (inflammation, extracellular matrix degradation, the apoptosis of endothelial and smooth muscle cells in the fibrous cap, larger lipid-rich core), the fibrous cap weakens, and this process may lead to plaque rupture and subsequent thrombosis with coronary occlusion [3,4]. Plaque rupture is the major cause of thrombosis in coronary vasculature and coronary occlusion [3]. Fewer than 25% of occlusions leading to acute coronary events develop in patients with advanced atherosclerotic plaques (more than 70% diameter) and more than two-thirds of acute events occur in subjects with non-significant lesions (<50% diameter) [6]. 

The exact molecular mechanisms of plaque erosion and rupture are not completely understood. It is, however, speculated to be a result of the degradation of type IV collagen connecting endothelial cells to the basal membrane by MMPs produced by activated macrophages [3,7].

CAD is a multifactorial disease with an important heritable component, and genetical factors have an integral role in the development of CAD and its complications (unstable angina, acute myocardial infarction, sudden cardiac death). Genetical factors (i.e., genetical polymorphisms) can be studied either by candidate gene approach or by genome-wide association studies (GWAS). The results of genetical studies (either candidate gene approach or GWAS) do not necessarily identify the causative single nucleotide polymorphisms (SNPs), and, therefore, additional functional studies are needed [8]. Recently, one of the largest genetic association studies of CAD was published [8]. They included more than half a million subjects: 122,733 cases and 424,528 controls [8]. They reported on the primary results of the meta-analysis of de novo GWAS data derived from the UK Biobank combined with existing data from CARDIoGRAMplusC4D [8]. At least 64 SNPs were found to be associated with CAD [8]. The contribution to the development of CAD of each identified locus is low, but the presence of multiple risk loci can have a profound effect. 

A disintegrin and metalloprotease with thrombospondin motif 7 (ADAMTS-7) gene is located on chromosome 15. It is composed of a signal peptide, prodomain, metalloproteinase domain, disintegrin-like domain, and thrombospondin type I motif. It is activated by the cleavage of signal peptide and pro-domain. In its active state, it is a proteolytic enzyme—ADAMTS7. It is produced and secreted by macrophages and is needed for the process of efferocytosis [9,10,11]. As a metalloproteinase, ADAMST7 is responsible for the degradation of cartilage oligomeric matrix protein (COMP), also known as thrombospondin-5 (TSP5), which is a component of vascular ECM and was also found to be present in atherosclerotic lesions [12]. The expression of ADAMTS-7 has been shown in human coronary arteries and human carotid atherosclerotic lesions [13,14]. A study found increased levels of ADAMTS-7 in lesions of symptomatic patients; furthermore, ADAMTS-7 levels correlated with the plaque characteristics of a vulnerable plaque phenotype [15]. Rat models show that the ADAMTS7 cleavage of TSP5 leads to vascular smooth muscle migration and promotes neointimal formation [16]. Therefore, it could promote the growth, rupture, and hemorrhage of the atherosclerotic plaque.

Genome-wide association studies (GWAS) showed that single nucleotide polymorphisms (SNPs) in ADAMTS7 were possibly correlated to coronary artery disease [17]. The association with CAD was found in five SNPs of ADAMTS7 (rs3825807, rs1994016, rs4380028, rs79265682, and rs28455815) [17]. A meta-analysis of the studies found the strongest association with CAD in ADAMTS7 SNP rs3825807 risk allele A, followed by rs4380028 (C vs. T allele) and rs1994016 (C vs. T allele) [17].

ADAMTS-7 is a potential target for the therapy of atherosclerosis. The knockdown of ADAMTS-7 decreases cartilage oligomeric matrix protein degradation [18], and the overexpression of COMP inhibits VSMC dedifferentiation [19]. Therefore, therapy that would decrease the expression of ADAMTS7 could potentially slow down the cleavage of COMP and decrease the migration of vascular smooth muscle cells and slow down the process of atherosclerosis.

The goal of the study was to investigate an association between the rs3825807 polymorphism of the *ADAMST7* and myocardial infarction in patients with type 2 diabetes. 

## 2. Materials and Methods

### 2.1. Subjects

In this retrospective cross-sectional case–control study, we studied 1590 unrelated White people who have had T2DM for at least 10 years. The participants were separated into two study groups: 463 subjects with MI and 1127 controls without a history of CAD, no ischemic changes during submaximal stress training, and no ECG signs of ischemic disease. The diagnosis of T2DM was carried out in accordance with the current criteria of the American Diabetes Association [20]. The history of MI in cases was diagnosed based on established universal standards [21]. Subjects with MI were incorporated in the study 1 to 9 months after the acute event. 

All participants who were included in the study were Slovenians of white ethnicity. Informed consent was signed by all participants. Afterwards, a detailed interview and physical examination were carried out. 

This research was funded by the National Research Agency of Slovenia ARRS, grant number P3-0326.

### 2.2. Ethical Statement

The study was approved by the National Medical Ethics Committee of Slovenia (number 0120-372/2017) and was performed in accordance with the Declaration of Helsinki.

### 2.3. Biochemical Analyses

We measured fasting glucose, total cholesterol, LDL cholesterol, HDL cholesterol, triglycerides (TG), and fasting glucose with the use of standard colorimetric assays on an automated biochemistry analyzer (Ektachem 250 Analyser, Eastman Kodak Company, Rochester, MN, USA). We calculated the LDL-c with the Friedewald formula. We defined hyperlipidemia as total cholesterol values over 5 mmol/L or TG over 2 mmol/L or as treatment with hypolipidemic medication.

Glycated hemoglobin (HbA1c) was estimated with high-performance liquid chromatography. We used the average value of three past HbA1c measurements. High-sensitivity C-reactive was measured with a Latex-enhanced immunonephelometric assay.

### 2.4. Genotyping

DNA was obtained from 100 µL of whole blood using a Qiagen isolation kit. The rs3825807 polymorphism of the ADAMST7 gene was genotyped by KBioscience Ltd. (LGC, Teddington, UK) using their own competitive allele-specific fluorescence-based PCR (KASPar) assay. Additional information is available at http://www.kbioscience.co.uk/, accessed on 20 November 2022. 

Genotyping the rs3825807 polymorphism of the ADAMST7 gene was performed by using KASPar chemistry (LGC Genomics, Middlesex, UK). The KASPar system employs a competitive allele-specific PCR combined with a FRET quenching reporter oligo. For each assay, three oligos are synthesized (two labeled allele-specific primers and one common reverse primer), and after amplification using genomic DNA as a template, the fluorophore signals are measured, and genotypes are determined. The KASPar system uses FAM and VIC fluorescence dyes to distinguish between genotypes and high ROX as a passive reference dye. PCR cycling under the manufacturers’ uniform conditions for measuring fluorescence levels and scoring genotypes by analyzing data for allele discrimination was used.

### 2.5. Statistical Analysis

Statistical analysis was performed with SPSS program version 19 (SPSS Inc., Chicago, IL, USA). Normally distributed continuous variables were presented as means ± standard deviation and in the case of asymmetrical distribution as median and interquartile range. The Kolmogorov–Smirnov test was used to examine the normality of the continuous variables. Normally distributed continuous variables were tested using unpaired Student’s *t*-test. Asymmetrically distributed variables were tested with the Mann–Whitney U-test. Discrete variables were compared with the Pearson χ^2^ test. The Pearson χ^2^ test was also used to determine if the genotype distribution deviates from Hardy–Weinberg equilibrium. Fisher’s Exact test was used to determine a significant relationship between two categorical variables if cells with expected frequencies lower than five were found in a contingency table. A stepwise multiple logistic regression was performed for all variables that showed significant deviation in univariate analysis. A *p*-value of <0.05 was considered to be of statistical significance.

## 3. Results

Clinical characteristics and biochemical parameters of cases (patients with MI) and controls are presented in Table 1. Both groups were well matched in body mass index, systolic blood pressure, fasting glucose, triglycerides, and incidence of drugs affecting the renin-angiotensin-aldosterone system. These cases were older, had a higher incidence of the male sex, higher total and LDL cholesterol, higher waist circumference, higher HbA1c, higher hs CRP, longer T2DM duration, and lower diastolic blood. Moreover, the cases had a higher incidence of statin use and a higher incidence of cerebrovascular stroke.

The distribution of genotype and allele frequencies of rs3825807 polymorphism of the ADAMST7 gene in subjects and controls is presented in Table 2. Cases and controls were put in Hardy–Weinberg equilibrium to determine their genotype distributions (cases: *p* = 0.584, controls: *p* = 0.06, Pearson χ^2^ test; respectively). The GG genotype was more frequent in the control group, and the AA genotype was more frequent in these cases. Moreover, the A allele was significantly more common in these cases than in the controls. 

Binary logistic regression was used to determine if rs3825807 polymorphism had an independent correlation with the progression of atherosclerosis after adjusting for age, waist circumference, diastolic blood pressure, HDL cholesterol, LDL cholesterol, TGS cholesterol, HbA1c, gender, smoking, and physical activity (Table 3). The results show a statistically significant association in two genetic models, codominant (OR 2.153; CI 1.215–3.968; *p* = 0.011) and recessive (OR 1.647; CI 1.120–2.407; *p* = 0.011). There was no correlation between the controls and the cases in the dominant genetic model (OR 1.673; CI 1.003–2.938; *p* = 0.059).

## 4. Discussion

In this study, we investigated an association between the rs3825807 of *ADAMTS7* and MI in Slovenian subjects with T2DM. Logistic regression analysis revealed an association between the A allele of the rs3825807 and MI in two genetic models (recessive and co-dominant).

The rupture of the plaque is the cause of the majority of acute coronary events [22]. Stable atherosclerotic plaques have a thick cap that protects the plaque from rupture, whereas unstable plaques have a big lipid-filled core covered by a thin fibrous cap [6,23]. The expression of ADAMTS-7 was shown in human coronary arteries and human carotid atherosclerotic lesions [13,14]. This study found increased ADAMTS7 levels in lesions of symptomatic patients. They also reported that the ADAMTS7 levels correlated with the plaque characteristics of an unstable and vulnerable plaque phenotype [15].

ADAMTS7 is located on chromosome 15q24 and is a protein-coding gene *ADAMTS7* [24]. It acts as a metalloproteinase and is produced by macrophages. ADAMTS proteases are first synthesized as inactive zymogens. Proprotein convertases (PCs) activate the proteases by removing the pro-domain [9,10,11]. ADAMTS7 acts as a protease on the cartilage oligomeric matrix protein also known as thrombospondin-5 (TSP5). TSP5 is an extracellular glycoprotein that is found in cartilage, tendon, vascular smooth muscle cells, and arteries. It plays a role in the adhesion and migration of vascular smooth muscle cells and was detected in normal and atherosclerotic arteries with immunohistochemistry [12]. The cleavage of TSP5 by ADAMTS7 contributes to vascular smooth muscle migration and promotes neointimal formation [16]. A study found the reduced formation of an atherosclerotic lesion in mice with ADAMTS7-deficiency [13]. In an atherosclerotic plaque, neovascularization promotes growth, rupture, and hemorrhage. Therefore, ADAMTS7 could be a potential therapeutic target to treat atherosclerosis.

A meta-analysis of the association of ADAMTS7 polymorphisms and the risk of developing CAD found the most consistent evidence with SNP rs3825807 risk allele A compared to the control G allele and the lesser, but still significant, evidence for rs4380028 (C vs. T allele) and rs1994016 (C vs. T allele) [17]. SNP rs3825807 G/G leads to a substitution of Ser-to-Pro at amino acid residue 214, which is the pro-domain of ADAMTS7 [25]. In vitro studies found that Ser-to-Pro substitution slowed down ADAMTS7 pro-domain cleavage [14]. A study revealed that in vitro assays of the GG genotype produced less of the cleaved ADAMTS7 pro-domain, less product of the ADAMTS7 substrate TSP5, and reduced migration ability of vascular smooth muscle cells [26]. Moreover, as higher levels of ADAMTS7 are correlated with a phenotype of a vulnerable plaque, this may be the reason for a higher probability of MI in patients with the AA genotype [15]. Consistent with these reports, our study found that the AA genotype was more common in cases who suffered MI. As genetic variants influence susceptibility to CVD, SNPs such as rs3825807 might be used to adjust individuals into risk categories more accurately in the future. However, currently, the use of GWAS discoveries as prognostic and therapeutic tools will need to be researched further to gain more knowledge on the connection between each locus and the phenotypes of the atherosclerotic lesion, the rupture of the plaque, and thrombosis.

ADAMTS7 is widely distributed in many tissues. ADAMTS7 also plays a role in the association with arthritis and disc disease [26]. Its main substrate TSP-5 is predominantly found in the ECM of cartilage. Increased concentrations of TSP-5 fragments and ADAMTS7 were measured in the synovial fluid and in the serum of patients with rheumatoid arthritis and osteoarthritis [26].

Many factors besides SNP play a role in the regulation of ADAMTS7. Proinflammatory cytokines, such as tumor necrosis factor-α (TNF-α), interleukin-1β (IL-1β), and platelet-derived growth factor (PDGF) induce the production of ADAMTS7 [14,27]. Reactive oxygen species (ROS) such as hydrogen peroxide, H_2_O_2_, also increase the expression of ADAMTS7. In contrast, the anti-inflammatory cytokine such as transforming growth factor β (TGF-β) inhibits the expression of ADAMTS7 [14,27]. Transcription factors, such as NF-κB and AP-1, which play a role in the migration and proliferation of vascular smooth muscle cells (VSMC), are also important for the regulation of ADAMTS7 expression in VSMC. In the promoter region of ADAMTS7, there are proinflammatory element binding sites, including NF-κB and AP-1 binding sites [27]. This evidence leads us to believe that the expression of ADAMTS7 is regulated at the level of transcription in response to inflammatory stimuli. Another factor in the regulation of ADAMTS7 is microRNA, miR-29a/b. MicroRNAs are short, non-coding RNAs that suppress the translation of the protein, and function in mRNA degradation. Additionally, miR-29a/b is downregulated by NF-κB, which also upregulates ADAMTS7 mRNA [28]. To sum up, the regulation of ADAMTS7 is a complex process that depends on the SNP ADAMTS7, proinflammatory and anti-inflammatory cytokines, ROS, transcription factors, and microRNA.

Additionally, ADAMTS7 has a stimulatory effect on the expression of inflammatory cytokines and proteases. It could also affect plaque stability through this mechanism. A positive feedback loop between ADAMTS7 and tumor necrosis factor-α (TNF-α) was observed in the cartilage [29]. Stimulating cartilage with ADAMTS7 induces the expression of MMP-1, -9, resulting in the degradation of collagen [29]. Furthermore, a study found a positive association between ADAMTS7 levels in the lesion and TNF-α, MMP-1, and MMP-9 [15]. The expression of MMP-1 and MMP-9 is associated with plaque rupture and localized in the shoulder region of human atherosclerotic plaques [30]. As ADAMTS7 is a part of an inflammatory feedback loop, ADAMTS7 may increase plaque vulnerability by promoting inflammation and matrix degradation through the induction of MMPs and other metalloproteinases belonging to the ADAMTS family.

ADAMTS7 has been localized in the shoulder region and the core of the plaque with the highest intensity [29]. It is possible that ADAMTS7, induced by an inflammatory stimulus, can be expressed by both SMCs and macrophages, and the expression pattern of ADAMTS7 is correlated with the cellular contents of the plaque. In fibrous lesions, ADAMTS7 may be expressed by SMCs and affects SMC migration; whereas, in atherosclerotic lesions with high lipid and inflammatory cells, its contents are mainly expressed in macrophages. This theory is supported by the fact that ADAMTS7 is expressed in several tissues including the heart, brain, muscle tissue, placenta, pancreas, liver, kidney, and testicles, and in several tissues, the expression of ADAMTS7 is increased by inflammatory stimuli [31]. For example, the angiotensin II treatment of mice which causes inflammation in the kidneys also induces renal ADAMTS7 expression [32].

In contrast to our findings, others failed to observe an association between the *ADAMTS7* rs3825807 polymorphism and cardiovascular disease [33]. Moreover, they failed to report an association between carotid intima-media thickness and the presence of carotid plaques in patients with rheumatoid arthritis [33]. In our study, we analyzed the association between the rs3825807 polymorphism and MI, which is caused by an acute coronary event, rather than subclinical atherosclerosis which is caused by the growth of plaque [22]. The AA genotype of the rs3825807 polymorphism leads to increased levels of ADAMTS7 [25,26]. Increased levels of ADAMTS7 are also found in lesions of symptomatic patients and are characteristic of vulnerable plaque phenotypes [15]. Therefore, the rs3825807 polymorphism may have a greater influence on the vulnerability of the plaque to rupture than on the growth of the plaque. Another study found that ADAMTS 7 has a greater influence on the growth of the plaque (i.e., angiographic CAD) rather than on the occurrence of early MI [34]. Additionally, in a postmortem study of human coronary arteries, Chan and co-workers demonstrated an association between the rs3825807 and reduced fibrous cap thickness, reduced fibrous cap-to-intima thickness, and a lower percentage area of α-actin in the intima [35]. They also reported that the rate of native vessel revascularization following enrolment was lower in those carrying the protective G allele [35]. Recently, it has been reported that the peptide vaccine against ADAMTS-7 ameliorates intimal hyperplasia in swine-stented coronary arteries, but without significant immune-related organ injuries [36]. They concluded that the ATS7 vaccine might be a novel atherosclerosis vaccine that also alleviates in-stent restenosis [36].

Our study has a few limitations. The first limitation is a selection bias due to the study design (i.e., retrospective versus prospective). Our study was a retrospective cross-sectional case–control study, and subjects with MI were enrolled in the study 1–9 months after the acute event (myocardial infarction). It is known that some patients die during the acute event, and due to that fact, it was not possible to be enrolled. Scientists in the field of population genetics are aware of the fact that prospective studies have advantages over retrospective studies. Moreover, a prospective design has been ranked higher in the hierarchy of evidence than a retrospective design [37]. The scientific community, on the other hand, appreciates retrospective studies since they are a convenient way of testing new hypotheses with existing data. A second limitation is the definition of the control group. Our study was searching for the association between the rs3825807 polymorphism and MI, with a control group defined by the absence of clinical signs of CAD. Therefore, the control subjects of our study could have had clinically silent CAD, which would affect the results. The presence of clinically silent CAD could be proven angiographically. Future studies should have a different design, with a control group that would be defined angiographically. 

## 5. Conclusions

In conclusion, our study showed that the AA genotype was associated with MI in subjects with type 2 diabetes; namely, it was more often present in the cases than in the controls. To summarize, the rs3825807 polymorphism of the ADAMTS7 gene could be used as a genetic marker for MI in patients with type 2 diabetes, and the AA genotype might be a genetic risk factor for MI.

## 6. Future Directions

Scientists in the field of population genetics are aware of the fact that prospective studies have advantages over retrospective studies, but at present, we are not planning such a design of the study. In the future, however, we plan to enroll new cases and control subjects to increase the sample size and power of the study. Cohort studies are very valuable due to the fact that they enable testing different interesting research hypotheses; however, ideally, they should be confirmed in prospective studies to confirm the causality of some factor (gene or gene marker). Moreover, we are also planning to add functional studies, since they are a good mode to test the functional significance of some biological markers (in our case, the rs3825807 polymorphism of the ADAMTS7 gene). Functional studies can be helpful to enlighten the understanding of the underlying pathogenetic mechanisms, and, eventually, this may lead to the translation of basic findings into new drug development. 

## Figures and Tables

**Table 1 genes-14-00508-t001:** Clinical characteristics of patients with T2DM with CAD and subjects with T2DM and without CAD.

	Cases (N = 463)	Control Subjects (N = 1127)	
Age (years)	66.45 ± 9.95	64.37 ± 9.21	<0.001
Systolic blood pressure (mm Hg)	147.97 ± 20.39	148.23 ± 20.85	0.829
Diastolic blood pressure (mm Hg)	81.68 ± 10.87	84.32 ± 10.38	<0.001
Gender			<0.001
Male	288 (62.2%)	573 (50.8%)	
Female	175 (37.8%)	554 (49.2%)	
Smoking (%)			<0.001
Never	367 (79.3%)	1010 (89.6%)	
Former smoker	23 (5.0%)	10 (0.9%)	
Active smoker	73 (15.8%)	107 (9.5%)	
Physical activity (3–4-times per week)	349 (75.4%)	759 (67.3%)	0.002
Body mass index (kg/m^2^)	29.56 ± 4.14	29.86 ± 4.80	0.245
Waist circumference (cm)	103.10 ± 12.23	107.78 ±12.48	<0.001
Fasting glucose (mmol/L)	8.84 ± 2.89	8.60 ± 2.64	0.173
HbA1c (%)	8.04 ± 1.32	5.95 ± 1.16	<0.001
hs CRP (mg/L)	2.30 (1.22–3.70)	1.90 (1.10–3.00)	<0.001
T2DM duration (years)	11.00 (6.00–20.00)	14.00 (10.00–20.00)	<0.001
Total cholesterol (mmol/L)	4.90 (4.00–5.90)	4.70 (4.00–5.60)	<0.001
HDL-cholesterol (mmol/L)	1.10 (0.90–1.30)	1.20 (1.00–1.40)	<0.001
LDL-cholesterol (mmol/L)	2.70 (2.10–3.70)	2.60 (2.10–3.30)	0.014
Triglycerides (mmol/L)	1.70 (1.20–2.60)	1.80 (1.20–2.50)	0.947
History of cerebrovascular stroke	59 (12.7%)	77 (6.8%)	<0.001
CABG history	123 (26.6%)	0 (0.0%)	<0.001
Stenting of coronary arteries	163 (35.2%)	0 (0.0%)	<0.001
Statin use	389 (84.0%)	763 (67.7%)	<0.001
Use of drugs affecting RAAS system	257 (55.5%)	592 (52.5%)	0.279

Abbreviations: HbA1c—glycated hemoglobin A1c; hsCRP—high-sensitivity C-reactive protein; CABG—coronary artery bypass grafting; RAAS—renin-angiotensin-aldosterone system. *p*-values in bold indicate statistical significance.

**Table 2 genes-14-00508-t002:** Distribution of rs3825807 polymorphism genotypes, alleles, dominant, and codominant genetic models within cases and controls, respectively.

	Cases (%) (N = 463)	Controls (%) (N = 1127)	*p* Value *	OR	*p* Value **
Genotypes_rs3825807			0.015		
AA genotype	154 (33.3%)	299 (26.5%)		1.55	0.007
AG genotype	231 (49.9%)	593 (52.6%)		1.17	0.29
GG genotype	78 (16.8%)	235 (20.9%)		Ref.	
Alleles			0.006		
A allele	539 (58.2%)	1191 (52.8%)		1.24	0.006
G allele	387 (41.8%)	1063 (47.2%)		Ref.	
Hardy–Weinberg equilibrium (*p* value)	0.5836	0.06124			
Dominant			0.068		
AA + AG	385 (83.2%)	892 (79.1%)		1.30	0.069
GG	78 (16.8%)	235 (20.9%)		Ref.	
Recessive			0.007		
AA	154 (33.3%)	299 (26.5%)		1.38	0.007
AG + GG	309 (66.7%)	828 (73.5%)		Ref.	

Abbreviation: OR—odds ratio. * *p* value from Chi-square; ** *p* value for OR.

**Table 3 genes-14-00508-t003:** Logistic regression model to explain the MI risk for the rs3825807 polymorphism of ADAMTS7.

Model	Adjusted OR (95% CI)	*p*-Value
co-dominant		
AA versus GG *	2.153 (1.215–3.968)	0.011
AG versus GG *	1.437 (0.836–2.583)	0.20
Dominant		
[AA + AG] versus GG *	1.673 (1.003–2.938)	0.059
Recessive		
AA versus [AG + GG] *	1.647 (1.120–2.407)	0.011

Abbreviations: OR—odds ratio; CI—confidence interval. * reference; Adjusted for: age, waist circumference, diastolic blood pressure, HDL-cholesterol, LDL-cholesterol, TGS-cholesterol, HbA1c; gender, smoking, physical activity.

## Data Availability

The data presented in this study are available on request from the corresponding author. The data are not publicly available due to ethical restrictions.

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
