# Peer review of "The rs3825807 Polymorphism of ADAMTS7 as a Potential Genetic Marker for Myocardial Infarction in Slovenian Subjects with Type 2 Diabetes Mellitus"

_genes, 2023, doi:10.3390/genes14020508_

Round 1
Reviewer 1 Report
This is an original research article, which aims to investigate the association between the rs3825807 polymorphism of ADAMTS7 and myocardial among patients with type 2 diabetes mellitus in a Slovenian cohort.
Generally, the topic is quite interesting, and the authors have in depth knowledge. They have used the appropriate methodology, study design, and adequate statistical analysis. The results are sufficiently well-presented, clear, and easy to understand, so as to reach safe and solid conclusions. Overall, the manuscript is well written and structured. Thus, I think it would make a nice addition to Genes as an original research article.
However, the following points should be generally considered, thus minor revision is demanded.
1. Line 55; Kindly analyze the role of ADAMTS7 in the pathophysiology of myocardial infraction and present also the relative clinical data.
2. Line 61; Kindly refer the abbreviation of COMP.
3. Discussion is quite short. The main study findings should be further interpreted in the part of discussion.
4. Line 185-188; Kindly comment about the conflicting data.
5. Future perspectives and clinical applications would be a nice addition to the manuscript, highlighting its scientific significance.
Author Response
Reviewer 1:
This is an original research article, which aims to investigate the association between the rs3825807 polymorphism of ADAMTS7 and myocardial among patients with type 2 diabetes mellitus in a Slovenian cohort.
Generally, the topic is quite interesting, and the authors have in depth knowledge. They have used the appropriate methodology, study design, and adequate statistical analysis. The results are sufficiently well-presented, clear, and easy to understand, so as to reach safe and solid conclusions. Overall, the manuscript is well written and structured. Thus, I think it would make a nice addition to Genes as an original research article.
However, the following points should be generally considered, thus minor revision is demanded.
- Line 55; Kindly analyze the role of ADAMTS7 in the pathophysiology of myocardial infraction and present also the relative clinical data.
This remark has been taken into consideration, and the information about the role of ADAMTS7 in the pathophysiology of myocardial infraction has been added in the Introduction section, paragraph 3, lines 6-8:
Expression of ADAMTS-7 has been shown in human coronary arteries and human carotid atherosclerotic lesions [b,16]. A study found increased levels of ADAMTS-7 in lesions of symptomatic patients, furthermore ADAMTS-7 levels correlated with plaque characteristics of a vulnerable plaque phenotype
- Line 61; Kindly refer the abbreviation of COMP
The abbreviation has been explained in the Introduction section, namely COMP is cartilage oligomeric matrix protein
- Discussion is quite short. The main study findings should be further interpreted in the part of discussion.
This remark has been taken into consideration, and we have added some additional information in the Discussion section (marked in yellow)
- Line 185-188; Kindly comment about the conflicting data.
This remark has been taken into consideration, and we have added relevant information in the Discussion section, paragraph 6, lines 3-9:
In contrast to our findings, others failed to observe an association between the ADAMTS7 rs3825807 polymorphism and cardiovascular disease [18]. Moreover, they failed to report an association between carotid intima-media thickness and the presence of carotid plaques in patients with rheumatoid arthritis [18]. In our study we analysed the association between the rs3825807 polymorphism and MI, which is caused by the acute coronary event, rather than subclinical atherosclerosis which is caused by the growth of the plaque [c]. AA genotype of the rs3825807 polymorphism leads to increased levels of ADAMTS7 [16,17]. Increased levels of ADAMTS7 are also found in lesions of symptomatic patients and are characteristic of vulnerable plaque phenotype [g]. Therefore the rs3825807 polymorphism may have a greater influence on the vulnerability of the plaque to rupture, than on the growth of the plaque. Another study found that ADAMTS 7 has a greater influence on the growth of the plaque (i. e. angiographic CAD), rather than on the occurrence of early MI [h].
- Future perspectives and clinical applications would be a nice addition to the manuscript, highlighting its scientific significance.
This remark has been taken into consideration, and we have added relevant information in the last paragraph of the Discussion section, lines 4-6 (marked in yellow):
Therefore, control subjects of our study could have clinically silent CAD, which would affect the results. Presence of clinically silent CAD could be proved angiographically. Future studies should have a different design, with a control group that would include CAD subject without MI.
Reviewer 2 Report
The manuscript needs some major revisions, as follow:
1. It is best to avoid using abbreviations and acronyms in the abstract.
2. It is necessary that the authors describe the details of the genetic test that was applied in this experiment.
3. The authors should have considered the genetic ancestry as the potential confounders in the statistical analysis.
4. Start your discussion with introducing your aim and then discuss the clinical significance of your experiment. Then discuss findings, followed by your study limitations and then conclusion.
5. In discussing your findings, compare and contrast them with other studies in the literature and develop arguments and hypotheses for your findings.
6. In addition to previous research aligning with yours, please critically discuss those in disagreement and develop arguments and hypothesize. You may also add recommendations.
7. Authors should report their comment regarding the fact that ESC recommends to avoid genetic testing in CVD apart from limiting cohorts of patients.
8. This is the hospital-based case-control study, which is prone to selection bias. I would recommend the authors’ thoughts about the potential selection bias in this study in the Discussion section.
9. The entire manuscript would benefit significantly from a grammatical revision.
Author Response
Reviewer 2:
The manuscript needs some major revisions, as follow:
1. It is best to avoid using abbreviations and acronyms in the abstract.
This remark has been taken into consideration, and there is only one abbreviation in the text (ADAMTS7)- but now explained.
- It is necessary that the authors describe the details of the genetic test that was applied in this experiment.
This remark has been taken into consideration, and we have described the details of genetic testing:
Genotyping the rs3825807 polymorphism of the ADAMST7 gene was performed by using KASPar chemistry (LGC Genomics, Middlesex, UK). KASPar system employs a competitive allele-specific PCR combined with a FRET quenching reporter oligo. For each assay three oligos are synthesized (two labelled allele specific primers and one common reverse primer) and after amplification using genomic DNA as a template the fluorophore signals are measured and genotypes are determined. The KASPar system uses FAM and VIC fluorescence dyes to distinguish between genotypes, and high ROX as a passive reference dye. PCR cycling under the manufacturers' uniform conditions for measuring fluorescence levels and scoring genotypes by analysing data for allele discrimination was used.
- The authors should have considered the genetic ancestry as the potential confounders in the statistical analysis.
This remark has been taken into consideration. The information regarding genetic ancestry is available in the first paragraph of the Material and methods section, subjects, lines 1 and 2.
In this retrospective cross-sectional case-control study we studied 1590 unrelated Caucasians with diagnosed T2DM lasting at least 10 years
- Start your discussion with introducing your aim and then discuss the clinical significance of your experiment. Then discuss findings, followed by your study limitations and then conclusion.
This remark has been taken into consideration, and we introduced the Discussion section with the aim of the study. We have also added the study limitations, and finished with the conclusion section - In discussing your findings, compare and contrast them with other studies in the literature and develop arguments and hypotheses for your findings.
This remark has been taken into consideration, and we have added relevant information in the Discussion section, paragraph 6, lines 3-9:
In contrast to our findings, others failed to observe an association between the ADAMTS7 rs3825807 polymorphism and cardiovascular disease [18]. Moreover, they failed to report an association between carotid intima-media thickness and the presence of carotid plaques in patients with rheumatoid arthritis [18]. In our study we analysed the association between the rs3825807 polymorphism and MI, which is caused by the acute coronary event, rather than subclinical atherosclerosis which is caused by the growth of the plaque [c]. AA genotype of the rs3825807 polymorphism leads to increased levels of ADAMTS7 [16,17]. Increased levels of ADAMTS7 are also found in lesions of symptomatic patients and are characteristic of vulnerable plaque phenotype [g]. Therefore the rs3825807 polymorphism may have a greater influence on the vulnerability of the plaque to rupture, than on the growth of the plaque. Another study found that ADAMTS 7 has a greater influence on the growth of the plaque (i. e. angiographic CAD), rather than on the occurrence of early MI [h].
in disagreement and develop arguments and hypothesize. You may also add recommendations.
This remark has been taken into consideration, and we have added relevant information in the Discussion section, paragraph 6, lines 3-9
In contrast to our findings, others failed to observe an association between the ADAMTS7 rs3825807 polymorphism and cardiovascular disease [18]. Moreover, they failed to report an association between carotid intima-media thickness and the presence of carotid plaques in patients with rheumatoid arthritis [18]. In our study we analysed the association between the rs3825807 polymorphism and MI, which is caused by the acute coronary event, rather than subclinical atherosclerosis which is caused by the growth of the plaque [c]. AA genotype of the rs3825807 polymorphism leads to increased levels of ADAMTS7 [16,17]. Increased levels of ADAMTS7 are also found in lesions of symptomatic patients and are characteristic of vulnerable plaque phenotype [g]. Therefore the rs3825807 polymorphism may have a greater influence on the vulnerability of the plaque to rupture, than on the growth of the plaque. Another study found that ADAMTS 7 has a greater influence on the growth of the plaque (i. e. angiographic CAD), rather than on the occurrence of early MI [h].
- Authors should report their comment regarding the fact that ESC recommends to avoid genetic testing in CVD apart from limiting cohorts of patients.
We are aware of the fact that genetic testing is not recommended in clinical practice for clinical risk assessment (ESC guidelines). Generally, different papers reporting associations of atherosclerotic coronary artery disease are research projects trying to demonstrate the importance of genetic factors (i.e. gene polymorphisms, epigenetics…) in the development of MI/CAD. In future they (DNA markers…) may or may not enter into the ESC guidelines.
- This is the hospital-based case-control study, which is prone to selection bias. I would recommend the authors’ thoughts about the potential selection bias in this study in the Discussion section.
This remark has been taken into consideration, and we have added additional information in the Discussion section, paragraph 7, lines 1-6:
Our study has a few limitations. Since it is a hospital-based case control study, it is prone to selection bias. A second limitation is the definition of the control group. Our study was searching for the association between the rs3825807 polymorphism and MI, with a control group defined by the absence of clinical signs of CAD. Therefore, control subjects of our study could have clinically silent CAD, which would affect the results. Presence of clinically silent CAD could be proved angiographically. Future studies should have a different design, with a control group that would include CAD subject without MI.
The entire manuscript would benefit significantly from a grammatical revision.
This remark has been taken into consideration, and the MS was checked by appropriate expert.
Round 2
Reviewer 2 Report
The authors should have considered the genetic ancestry as the potential confounders in the statistical analysis.
Unfortunately, this recommendation has not been addressed.
Manuscript needs statistical data for genetic ancestry as a potential confounder.
Author Response
This remark has been taken into consideration, and we have added some additional information in the text.
We have added some additional information that was demanded.
- The structure of our original paper was already organized as demanded, however we added another suggested paragraph of the original paper, Future directions.
- Moreover, we added some text (marked in blue) to reach almost 4000 words in our original research paper.
